# CD4^+^ T Cells in Chronic Hepatitis B and T Cell-Directed Immunotherapy

**DOI:** 10.3390/cells10051114

**Published:** 2021-05-06

**Authors:** Sonja I. Buschow, Diahann T. S. L. Jansen

**Affiliations:** Department of Gastroenterology and Hepatology, Erasmus MC University Medical Center, 3015 GD Rotterdam, The Netherlands

**Keywords:** CD4^+^ T cells, hepatitis B virus, immunotherapy

## Abstract

The impaired T cell responses observed in chronic hepatitis B (HBV) patients are considered to contribute to the chronicity of the infection. Research on this impairment has been focused on CD8^+^ T cells because of their cytotoxic effector function; however, CD4^+^ T cells are crucial in the proper development of these long-lasting effector CD8^+^ T cells. In this review, we summarize what is known about CD4^+^ T cells in chronic HBV infection and discuss the importance and opportunities of including CD4^+^ T cells in T cell-directed immunotherapeutic strategies to cure chronic HBV.

## 1. Introduction

The hepatitis B virus (HBV) specifically infects hepatocytes, leading to acute or persistent liver infection. The majority of infected adults can resolve the infection, but 5–10% develop chronic hepatitis, which can lead to cirrhosis, hepatocellular carcinoma (HCC) and death [1]. Current therapy consists of suppression of viral replication by lifelong use of nucleos(t)ide analogues (NA) which reduces, but does not abrogate HCC risk. The viral HBV genome is formed as covalently closed circular DNA (cccDNA) which encodes for: three forms of the HBV surface antigen (HBsAg), the viral capsid-forming core (HBcAg) and its secreted form called the e antigen (HBeAg), viral polymerase and the non-structural X protein. In addition to complete viral particles, large amounts of HBeAg and HBsAg are secreted by infected hepatocytes, presumably as a decoy for the immune system [2]. HBeAg and HBsAg can be detected in serum and are used to define the different phases of chronic infection together with serum HBV DNA levels and markers of liver disease (i.e., serum alanine aminotransferase (ALT) levels and fibrosis scoring) [1].

Most patients start with a non-inflammatory phase with high levels of HBV DNA and HBsAg, the presence of HBeAg and normal ALT levels (HBeAg-positive chronic infection; EPCI), followed by an inflammatory phase with increased ALT levels, fluctuating HBV DNA levels and the presence of HBeAg (HBeAg-positive chronic hepatitis; EPCH), while some patients remain in the EPCI phase. The EPCH phase may mount into partial viral immune control with loss of HBeAg expression, often accompanied by generation of antibody responses to HBeAg, undetectable or low HBV DNA levels and normalization of ALT levels (HBeAg-negative chronic infection; ENCI). Patients may stay in this phase for years, but viral replication could return, leading to high or fluctuating HBV DNA and ALT levels and liver disease (HBeAg-negative chronic hepatitis; ENCH) [1].

Viral clearance is largely dependent on the adaptive immune response including effector T cells and neutralizing antibodies, which is illustrated by the observation that pre-existing HBV immunity can clear established chronic HBV infection (cHBV) in the context of bone marrow and liver transplantation [3,4,5,6]. Furthermore, studies in chimpanzees demonstrated that CD8^+^ T cells are essential effector cells responsible for viral clearance during acute infection, as depletion of CD8^+^ T cells during the clearing phase of the infection resulted in viral persistence [7]. The T cell response in acute infection is robust, long-lasting and directed against the nucleocapsid, surface, polymerase and X proteins of the virus [8,9,10,11,12,13,14,15,16]. This multi-specificity is associated with resolution [17,18]. However, in chronically infected patients, the T cell response is weak and narrow, leading to viral persistence [10,12,15,19]. Activating, boosting or introducing HBV-specific T cell responses is therefore an interesting therapeutic strategy for chronic infection. Different approaches including therapeutic vaccination have been developed over the past decades, but thus far with limited success, as we and others have recently reviewed in detail [20,21,22]. Focus has been on CD8^+^ T cells because of their cytotoxic effector function; however, CD4^+^ T cells may be equally important for viral clearance because they are needed for the development of optimal effector CD8^+^ T cells and for the generation and maintenance of functional memory CD8^+^ T cells, amongst others [23,24]. In this review, we will zoom in on what is known about CD4^+^ T cells in acute and chronic hepatitis B infection and the importance and opportunities of including CD4^+^ T cells in antigen-specific T cell-directed therapy to cure cHBV.

## 2. CD4^+^ T Cells in Viral Infections

In general, in response to viral infection, several subsets of antigen-experienced CD4^+^ T cells are generated from naive CD4^+^ T cells to aid virus elimination. CD4^+^ T cells are mostly known for their help to other cells of the adaptive immune system rather than for exerting effector functions themselves, although exceptions exist (reviewed by Swain et al. [23] and below). The best-studied helper function of CD4^+^ T cells is the promotion of antibody production by B cells through germinal center formation, isotype switching and affinity maturation [23] (Figure 1). Follicular helper T (Tfh) cells are the specialized CD4^+^ T cell subset that help B cells through cell-cell interactions, mostly via CD40L-CD40 interactions, and release of cytokines (reviewed by Crotty [25]). CD4^+^ T cells also promote effector and memory CD8^+^ T cell responses through licensing of antigen presenting cells (APC) via CD40L-CD40 interactions and cytokine secretion [23,26,27,28,29] (Figure 1). The help signals from CD4^+^ T cells are transferred to CD8^+^ T cells by lymph node-resident conventional type 1 dendritic cells (cDC1) that increase their antigen presentation capability and expression of co-stimulatory molecules and cytokines in response to CD40 triggering (reviewed by Borst et al. [24] and [30,31]). The type I interferon, IL-12 and IL-15 secreted by CD40 activated cDC1 act directly on cytokine receptors on the CD8^+^ T cells, while the costimulatory receptors CD80/CD86 and CD70 expressed on CD40 activated cDC1 interact with CD28 and CD27, respectively [24]. Together with the antigen-specific T cell receptor (TCR) triggering delivered by HLA-peptide complexes on cDC1, these signals drive differentiation of the CD8^+^ T cells into potent long-lasting cytotoxic T cells (CTL) [24]. IL-2 and IL-21 produced by the CD4^+^ T cells also support the CTL response directly [24]. Furthermore, effector CD4^+^ T cells can migrate to the site of infection and protect against viral pathogens through the local production of cytokines (IFNγ and TNFα) and direct cytolytic activity mediated by perforin and FAS (Figure 1) [23]. In contrast, IL-10 production by CD4^+^ T cells regulates immunopathology [23,32].

CD4^+^ T cells generated in response to viral infection are mainly of the T helper 1 (Th1) phenotype and produce large amounts of IFNγ, TNFα and IL-2 and express T-bet when exposed to IL-12 and type I interferons. IFNγ and TNFα have important anti-viral effects, as they can enhance the antiviral activity of macrophages (by stimulation of nitric oxide production), induce resistance to virus in neighboring cells and increase expression of HLA molecules on infected cells [33]. However, Th2 and Th17 cells, characterized by the respective secretion of IL-4 and IL-17 have also been reported to some extent in viral infections [34,35,36]. While Th2 cells are needed to clear extracellular parasites, bacteria and allergens, their dominance during viral infections is thought to be unfavorable, since Th2 cytokines can counteract the Th1 effect [34,37,38]. Nonetheless, from several studies, it became clear that different Th subsets can drive distinct antibody class switching, indicating that both Th subsets are responsible for the broad range of protective antibody isotypes found in patients with anti-viral immunity; Th1 enhances IgG2a class switching in mice which is equivalent to IgG1 in humans, while Th2 are instead involved in IgG1 production (equivalent to IgG2a in humans) [25,39,40]. Finally, Th17 mainly drive inflammation, but their ability to recruit neutrophils could offer viral protection and besides, via IL-21, Th17 may sustain the anti-viral CD8^+^ T cell response [41,42]. Interestingly, substantial plasticity exists within Th subsets in vivo, especially during responses to pathogens, indicating the relation between subtypes and viral clearance is highly complex. Th2 can acquire a mixed Th1/Th2 phenotype and Th17 can be reprogrammed to Th1 under influence of IL-12 and type I interferons [43,44,45,46]. Moreover, IL-10 can be produced by a subset of the different T helper subsets. So, viral clearance likely depends on multiple Th subsets that together are capable of providing help to B cells and cytotoxic CD8^+^ T cells, mediating direct antiviral effector functions and regulating immunopathology. In the following paragraphs, we discuss CD4^+^ T cells and the different Th subsets and their (dys)function in HBV.

## 3. CD4^+^ T Cells in Acute Hepatitis B

The seminal chimpanzee studies defining the important role of CD8^+^ T cells in viral clearance also addressed the role of CD4^+^ T cells. Depletion of CD4^+^ T cells at a later stage after infection did not change its course, in contrast to the observed viral persistence upon CD8^+^ T cell depletion at this same moment [7]. However, depletion of CD4^+^ T cells early on, before viral spread, did result in persistent infection [47]. Furthermore, in murine studies, CD4^+^ T cell depletion also resulted in reduced and impaired virus-specific CD8^+^ T cell responses, leading to chronic liver infection [48]. These observations show that CD4^+^ T cells are most likely not important as effector cells in clearing acute HBV, but do indicate a critical role for CD4^+^ T cells in HBV infection control. Of note, these observations do not exclude an effector role in cHBV. A mechanism of action to explain the requirement for CD4^+^ T cells at the early stage of infection may be that CD4^+^ T cells are needed to help CD8^+^ T cells develop into functional effector cells and/or to aid the production of virus neutralizing antibodies. More support for a role for CD4^+^ T cells in clearance and chronicity comes from the association of HLA class II alleles with HBV clearance or persistence. A meta-analysis showed that HLA-DR*04 and HLA-DR*13 alleles were significantly associated with HBV clearance, while HLA-DR*03 and HLA-DR*07 were associated with persistence [49]. Also HLA-DQ and HLA-DP alleles have been associated with persistent HBV infection and response to therapy [50,51]. Furthermore, a heterozygosity advantage has been reported to clear HBV infection for HLA II, but not HLA I [18].

In the natural course of infection, CD4^+^ T cells appear in the blood simultaneously with CD8^+^ T cells between 7 to 10 weeks after infection and before symptoms develop [52,53]. They recognize epitopes derived from the core, polymerase, surface and x proteins, thus encompassing a multi-specific CD4^+^ T cells response, similar to the CD8^+^ T cells response [8,9,12,15,54]. Studies focusing on the function of CD4^+^ T cells in HBV reported predominant secretion of IFNγ, but no or low IL-4 and IL-5, suggesting a dominant Th1-cytokine profile by HBcAg-specific CD4^+^ T cells [55,56]. A more recent study investigating the production of the anti-viral Th1 cytokines IFNγ, IL-2 and TNFα described predominant production of IFNγ as well as some IL-2, but no TNFα by core-, surface-, polymerase- and x-specific CD4^+^ T cells [15]. The observed Th1 profile is in line with the murine studies described above in which Th1 induces IgG1 production, since neutralizing anti-HBs antibodies are of the IgG1 and IgG3 subclass in resolved individuals [57]. Interestingly, increased CD4^+^ T cell responses to HBcAg/HBeAg is associated with HBeAg loss and/or emergence of anti-HBe antibodies in acute hepatitis B infection, suggesting that HBcAg/HBeAg-specific CD4^+^ T cells are essential for viral elimination [58].

So, in acute self-limiting infection, a strong, multi-specific Th1 CD4^+^ T cell response is elicited that has a crucial role in HBV control.

## 4. CD4^+^ T Cells in Chronic Hepatitis B

Comparable to HBV-specific CD8^+^ T cells, the CD4^+^ T cell response in cHBV is more narrow as compared to acute clearing infection, demonstrated by decreased recognition of epitopes [54]. Furthermore (and also similar to HBV-specific CD8^+^ T cells), HBV-specific CD4^+^ T cells are thought to be exhausted because of their high expression of inhibitory molecules PD-1, CTLA-4 and LAG-3 and show loss of function by reduced secretion of cytokines and reduced in vitro proliferation capacity (Figure 2A) [9,12,56,59,60]. Several studies have reported decreased secretion of Th1 cytokines IFNγ, IL-2 and TNFα by CD4^+^ T cells in cHBV compared to acute infection indicating a dysfunctional Th1 response in cHBV [54,56,59,60,61,62]. Of note, the frequency of HBV-specific CD4^+^ T cells in cHBV is lower compared to acute infection and according to some studies even hardly detectable directly ex vivo [58,63]. However, after 10-day in vitro HBV-peptide stimulation IFNγ and TNFα production could be observed, indicating remaining cytokine producing capacity of HBV-specific CD4^+^ T cells upon restimulation [63,64]. Interestingly and important for therapy development, during a hepatitis B flare, TNFα- and IFNγ-producing HBV-specific CD4^+^ T cells could be detected, further indicating that the CD4^+^ T cells still poses the capability of type I cytokine production when re-activated [64].

A skewed Th1/Th2 ratio towards Th2 cells is considered unfavorable for clearance of viral infections [34,38]. Skewing Th responses into the Th2 direction could therefore be a successful immune evasion strategy for a virus as this will lead to a Th response that is not capable of clearing the virus, giving the virus a survival advantage. Experimental evidence argues that the secreted HBeAg may fulfill this task for HBV [65]. Mice were immunized with either HBcAg or HBeAg and the T helper response was determined against both HBcAg and HBeAg. In vivo HBcAg-primed Th cells mostly produced IFNγ and IL-2 and displayed a Th1 phenotype when exposed to HBcAg or HBeAg. HBeAg-primed T cells, in contrast, produced predominantly IL-4 in response to both antigens and were therefore Th2 cells. In line with these observed Th subsets, immunization with HBcAg resulted in mostly IgG2 antibodies while HBeAg immunization lead to IgG1 production [65]. Using a HBeAg-transgenic mouse model, the same group also reported depletion of HBeAg- and HBcAg-specific Th1 CD4^+^ T cells related to HBeAg expression via fibroblast-associated (FAS)-mediated apoptosis [66]. Further evidence for a Th2 skewing effect of HBeAg was obtained from a study on therapeutic vaccination that demonstrated skewing of vaccine responsive CD4^+^ T cells to IL-5 producing Th2 cells in HBeAg-positive patients, while vaccine responsive CD4^+^ T cells derived from healthy controls produced significantly more IL-12 and IFNγ [67]. Thus, circulating HBeAg may induce Th2 cells and deplete Th1 cells, thereby downregulating antiviral clearance mechanisms.

Th17 cells are associated with inflammation and have been described to be increased in cHBV. Compared to healthy controls, cHBV patients exhibit a higher percentage of Th17 cells in their blood detected by intracellular cytokine staining after non-specific stimulation [61,62,68]. Th17 frequency positively correlated with both plasma HBV DNA load and serum ALT levels [68]. Not only peripheral, but also intrahepatic Th17 cells are augmented in cHBV [68]. Because of the potential of IL-17 to activate mDCs and monocytes to release inflammatory cytokines in vitro and recruit neutrophils to the site of infection, an excessive Th17 response could be involved in liver injury during chronic infection [42,68,69].

Not only an increased frequency of Th17 cells, but also an increased frequency of regulatory T cells (Tregs) in the peripheral blood of cHBV patients has been reported [70,71]. These Tregs were defined as CD4^+^CD25^+^CD45RO^+^CTLA-4^+^ T cells since flow cytometric detection of transcription factor Foxp3 was not yet routinely used to identify Tregs 15 years ago. However, also higher expression of Foxp3 RNA was detected in cHBV patients [70,72]. In line with these results, a more recent study identified an increase of Foxp3^+^CD127^-^ Tregs in the circulation of cHBV patients [59]. Functionally, cHBV Tregs are capable of inhibiting HBV-specific Th1 and Tfh responses, indicating that Tregs can contribute to an inadequate T cell response against the virus [70,73].

Production of anti-HBe and anti-HBs by B cells is important for HBV clearance and is mediated by Tfh in part through their production of IL-21. However, in cHBV, IL-21 production in response to HBsAg peptides was decreased compared to acute infection [74]. Yet, an increased frequency of circulating Tfh was found and associated with HBeAg seroconversion as higher levels were detected in cHBV patients with HBeAg loss [74,75,76]. Furthermore, the Tfh from HBeAg seroconverters produced higher levels of IL-21 and harbored augmented frequencies of anti-HBe antibodies secreting B cells in vitro [75]. Conversely, a CD25^+^Foxp3^+^ Treg-like subset within the CD4^+^CXCR5^+^ circulating Tfh cells has been reported to be enriched in cHBV and to promote regulatory B cell functions which could hamper antibody mediated HBV clearance [73].

In summary, dysfunction of CD4^+^ T cells in cHBV is not confined to one subset, but all CD4^+^ T cell subsets are affected by the chronic presence of HBV (Figure 2B). Therefore, the CD4^+^ T cell response is likely suboptimal for viral clearance, contributing to poor CD8^+^ T cell and B cell responses and unwanted inflammation and liver damage.

## 5. CD4^+^ T Cells in Past Clinical Studies Testing T Cell-Directed Therapies

Over the past decades, several T cell-directed strategies to restore dysfunctional and/or obtain T cell responses have been developed, including immune checkpoint blockade (ICB), adoptive T cell therapy and therapeutic vaccination (TV) (reviewed in Lang et al. and Bertoletti et al. [77,78]). ICB is acting on existing T cell responses, whereas TV can boost existing responses and induce de novo responses from naïve T cells. Adoptive T cell therapy with engineered T cells replaces the dysfunctional HBV-specific CD8^+^ T cells response altogether. TV can be based on HBV-derived peptides or proteins, nucleic acids encoding HBV antigens or in vitro loaded monocyte derived dendritic cells. We recently reviewed past clinical trials testing these different types of TV in cHBV and discussed potential reasons for their limited success [20]. Most of the trials mainly focused on clinical effects and did not monitor the T cell response in detail and used assays that are not discriminative for CD4^+^ or CD8^+^ T cells responses such as IFNγ ELISpot. Yet, a few clinical trials provided relevant insights into the importance of including CD4^+^ T cells in TV strategies.

Theradigm-HBV is a peptide-based TV consisting of the well-described c18–27 epitope combined with the T helper epitope Tetanus toxoid (TT) 830–843 [79]. Despite a lack of clinical effect, this study demonstrated that CD4^+^ T cells are important for the induction of primary CD8^+^ T cell responses and their longevity [67,79]. Furthermore, two different TV composed of HBsAg were capable of inducing HBsAg-specific CD4^+^ T cells that produced IFNγ and IL-2 (and no IL-10) identified by in vitro proliferative response which disappeared upon CD4 depletion [80,81]. Interestingly, no CD8^+^ T cell responses were induced by these TV. The DNA vaccine pCMV-S2.S encoding pre-S2 and S also induced IFNγ-producing (Th1) CD4^+^ T cells (no other cytokines were tested) [82,83]. HBsAg-specific CD4^+^ T cells are not the only ones that can be induced by TV, as DNA vaccine HB-100 was capable of inducing surface-, core- and polymerase-specific CD4^+^ T cells that produce IFNγ (no other cytokines were tested) [84]. Finally, GS-4774 which is composed of whole yeast cells expressing HBsAg, HBcAg and X induced effective CD8^+^ T cell responses, but lacked clinical effect. This disappointing result may be explained by the absence of TV-induced CD4^+^ T cells, as exclusively CD8^+^ T cell responses were triggered [85,86].

Even though none of these TV lead to functional cure (i.e., undetectable HBV DNA and loss of HBsAg +/− anti-HBs antibodies) reduction of serum HBV DNA and even HBsAg loss was only observed in patients with the strongest CD4^+^ T cell response suggesting that CD4^+^ T cells are important in controlling viremia upon TV [81,84]. Knowledge on the induction/reactivation of CD4^+^ T cells by TV strategies is, however, far from complete, but these past studies clearly suggest that HBV-specific CD4^+^ T cells producing a Th1 cytokine profile can be induced by TV and that CD4^+^ T cells are important for induction of CD8^+^ T cell responses and controlling HBV replication. The road to viral clearance or control may lie with TV that can effectively trigger both Th1 and CD8^+^ T cells.

## 6. Harnessing CD4^+^ T Cells with Future Therapeutic Vaccination

As we outlined, available data on HBV and current knowledge on the important role of CD4^+^ T cells for effective CTL and B cell activation suggest that TV strategies for cHBV may strongly benefit from approaches with a high capacity to induce and/or re-activate HBV-specific CD4^+^ T cells. Such CD4^+^ T cells may not only help the induction of effective anti-viral CD8^+^ T cells, but potentially also can restore impaired existing HBV-specific CD8^+^ T cells (if not terminally exhausted). Due to the stealth nature of HBV and therefore poor activation of dendritic cells and CD4^+^ T cells, HBV-specific CD8^+^ T cells in cHBV have potentially been activated in the absence of CD4^+^ T cell help. This “helpless” CD8^+^ T cell priming causes impaired effector function, which can manifest as an exhausted phenotype. Helpless CD8^+^ T cells express higher levels of inhibitory molecules, lower levels of cytotoxic effector molecules and show impaired migratory capacity compared to CD8^+^ T cells that did receive CD4^+^ T cell help [87]. Surrogate CD4^+^ T cell help signals in the form of CD27 or CD40 agonistic antibodies, offered after priming, can improve the migratory potential and cytotoxic effector molecules and decrease the levels of inhibitory molecules. Surrogate help has been shown to also be able to restore cytotoxic functions in a murine model for cHBV [87,88]. So, inducing and/or activating CD4^+^ T cells in addition to CD8^+^ T cells by TV may induce better effector CD8^+^ T cells and may even restore existing impaired CD8^+^ T cells in addition to reactivation of anergic HBV-specific B cells [89]. Exploiting already existing memory HBV-specific CD4^+^ T cells would have preference over inducing primary effector CD4^+^ T cells, as secondary effector cells arising from memory CD4^+^ T cells show greater expansion and a higher capacity to secrete multiple cytokines (provided these are not irreversibly exhausted) [23].

When aiming to promote HBV-directed CD4^+^ T cell responses, the most important factor is the induction of IFNγ-producing Th1-skewed CD4^+^ T cells, as these cells have been associated with viral clearance [64]. Since HBeAg has been described to induce IL-4 producing Th2 cells [65], induction of IFNγ-producing CD4^+^ T cells may be optimal under HBeAg free conditions. These can be achieved either by selecting HBeAg-negative cHBV patients or reducing HBeAg levels by antiviral treatment and/or siRNA. This notion is supported by (1) the increased frequencies of HBV-specific IFNγ-producing CD4^+^ T cells observed in patients with HBsAg loss (following HBeAg loss) upon nucleos(t)ide analogue (NA) or IFNα treatment, and (2) the shift from liver damage inducing TNF-producing to viral clearance inducing IFNγ producing CD4^+^ T cells in HBeAg-negative patients [15,64,90]. Another approach of warranting induction of IFNγ production by CD4^+^ T cells is ensuring that IL-12 (and IL-18) are present during their activation in vivo, which can be achieved by including the correct adjuvant in the TV strategy, such as TLR3, TLR4, TLR7 or TLR9 ligands for peptide/protein-based TV or IL-12 coding DNA/RNA for DNA/RNA vaccines [91,92]. Importantly, IL-12 may also induce a shift from Th2 to Th1 in existing Th responses, thereby correcting the skewed Th1/Th2 balance in cHBV [65].

Given the limited success of TV in the clinic, TV alone will probably not be sufficient to reach a functional cure. Based on the evidence outlined above, we believe TV that include epitopes to trigger both Th1 CD4^+^ and CD8^+^ T cells, such as synthetic long peptides (SLP) or DNA/RNA vaccines, will have greater chances of success than vaccines based on CD8 epitopes alone. However, combination of TV with T cell enhancing drugs, like checkpoint inhibitors or T cell metabolism modifying drugs, might ultimately be needed for success. Targeting PD1/PDL1 holds great promise as blockade of the PD1 pathway has shown to increase IFNγ, IL-2 and TNFα production by HBV-specific CD4^+^ T cells [60,63,93]. While aiming to promote CD4^+^ IFNγ responses by TV seems justified, it is less clear whether CD4^+^ TNFα responses are also beneficial. TNFα holds the capacity to inhibit the suppressive effect of Tregs on the HBV-specific T cell response [71]. However, TNFα-producing CD4^+^ T cells have been found to be associated with liver damage [64].

We recently proposed a treatment strategy including a stop of NA treatment serving as a natural booster [20]. Studies of CD4^+^ T cells in patients enduring a viral flare provide support for this approach. Wang et al. reported that the patients who experienced HBeAg or HBeAg and HBsAg loss shortly after the flare showed higher frequency and dominance of IFNγ-producing CD4^+^ T cells compared to those with HBeAg persistence, supporting NA stop as part of the treatment strategy [64]. Interestingly, these IFNγ-producing CD4^+^ T cells also produce IL-21, which is not only essential for B cells but also supports CD8^+^ T cells to avoid deletion, maintain immunity and resolve persistent infection [94].

## 7. Concluding Remarks

We here outlined that because CD4^+^ T cells are crucial for the development of strong CD8^+^ T cells with optimal effector functions, both CD8^+^ and CD4^+^ T cells should be targeted in TV strategies to cure cHBV. In acute HBV infection, both the multi-specific CD8^+^ and CD4^+^ T cell responses with a dominant Th1 cytokine profile are likely responsible for clearance of the virus together with neutralizing antibodies. Therefore, TV aims to induce multi-specific CD8^+^ and Th1 CD4^+^ T cells, as they possess proven virus killing capacities. To reach this aim, we envision a stepwise approach. First, HBeAg levels may need to be reduced to remove the Th2 skewing environment by antiviral treatment and/or siRNA-based therapy. When viral load/HBeAg levels are stably low, HBV-specific CD4^+^ and CD8^+^ T cells can be induced and/or boosted using TV. SLP or DNA/RNA vaccines are specifically equipped for this task, as both CD4 and CD8 epitopes can be included that are also physically linked to each other to ensure presentation by the same cDC1 in the lymph node. To warrant Th1 induction and potentially reverse Th2 skewing, an adjuvant capable of inducing Th1 skewing/IL-12 production is desired. To reach optimal (CD4^+^) T cell function, TV can be combined with T cell enhancing drugs (TED) such as anti-PD1 or metabolism modifying drugs. Timing of TED will depend on the type of TED as anti-PD1, for instance, will be most beneficial after (first) TV administration, while other types of TED should be administered before or simultaneously with TV. As discussed previously ([20] and above), NA treatment stop can serve as a natural booster by increasing viral antigen presentation that will boost HBV-specific T cells in situ to clear remaining infected hepatocytes. This NA stop must be well-timed and only performed when high numbers of good quality HBV-specific T cells are achieved. To obtain this optimal time window for NA stop, another round or a different type of TED (TED boost) could be provided to expand existing HBV-specific (CD4^+^) T cells (Figure 3).

Of note, studies on HBV-specific CD4^+^ T cells directly ex vivo are few and rather limited rendering our knowledge of these cells incomplete. Most studies have been performed in the 90s using proliferation and ELISpot assays that may have missed significant parts of the response [95,96]. Direct methods such as those exploiting peptide-MHC-multimers in combination with high dimensional phenotyping by flow cytometry, mass cytometry or single cell RNA seq are more sensitive and quantitative, but require knowledge of epitopes and HLA type. Furthermore, the instability, complexity and variety of MHC class II molecules pose great challenges. Nonetheless, now several HLA class II multimers, including DRB1*01:01_core 61–80_, are commercially available leading to more extensive knowledge on core-specific CD4^+^ T cells compared to the other specificities [93,97]. Future studies are warranted to elucidate the impairment of HBV-specific CD4^+^ T cells in cHBV. Outstanding questions are whether CD4^+^ T cells of different specificities, like their CD8^+^ counterparts, may also be differentially dysfunctional and if HBV-specific CD4^+^ T cells also harbor an altered metabolism, as has been reported for HBV-specific CD8^+^ T cells [19,98,99,100,101].

In conclusion, CD4^+^ T cells are highly important for the induction of an effective adaptive response to clear HBV. Therefore, we believe CD4^+^ T cells should be included in TV strategies to cure cHBV. Nonetheless, our knowledge on HBV-specific CD4^+^ T cells is incomplete and further studies using the most recent methodological developments are necessary to fully understand their impairment in cHBV and deploy them to their fullest potential in TV strategies.

## Figures and Tables

**Figure 1 cells-10-01114-f001:**
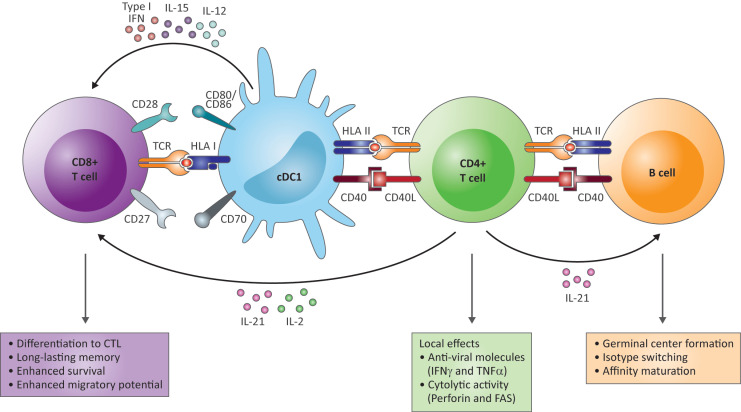
The functions of CD4^+^ T cells in the adaptive immune response to HBV. In response to viral infection, CD4^+^ T cells promote antibody production by B cells through germinal center formation, isotype switching and affinity maturation via CD40L-CD40 interactions and release of IL-21 (right and orange box). CD4^+^ T cells also support the cytotoxic T cell (CTL) response through conventional type 1 dendritic cells (cDC1) (left). CD40L-CD40 interactions result in increased expression of costimulatory receptors CD80/CD86 and CD70 on CD40-activated cDC1 that interact with CD28 and CD27, respectively, on the CD8^+^ T cells and in secretion of type I interferon, IL-12 and IL-15. These signals, together with T cell receptor (TCR) triggering via HLA I peptide complexes and IL-2 and IL-21 produced by the CD4^+^ T cells, induce differentiation into CTL with enhanced activity, enhanced survival and migratory potential and long-lasting memory (left and purple box). Additionally, effector CD4^+^ T cells can migrate to the site of infection and protect against viral pathogens through the local production of cytokines (IFNγ and TNFα) and direct cytolytic activity mediated by perforin and FAS (green box). This figure is an integrated extract of the most important roles of CD4^+^ T cells in the adaptive immune response, as reviewed in Swain et al. [23] and Borst et al. [24].

**Figure 2 cells-10-01114-f002:**
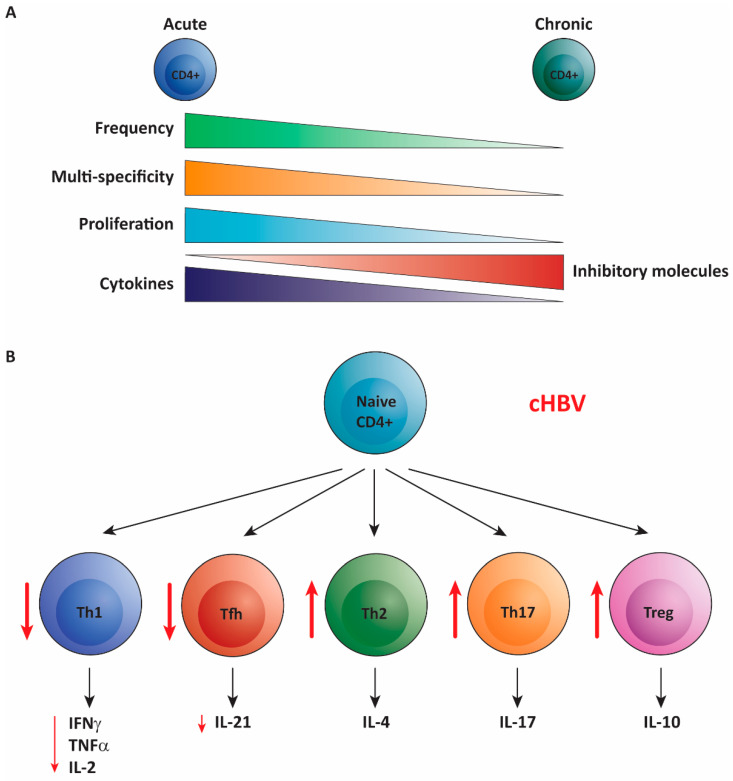
Overview of the (dys)function of HBV-specific CD4^+^ T cells in chronic hepatitis B infection. (**A**) Compared to HBV-specific CD4^+^ T cells observed in acute hepatitis B infection (**left**) the HBV-specific CD4^+^ T cells in chronically infected individuals (cHBV) are low in frequency, recognize decreased number of epitopes, exhibit lower proliferative potential, produce less cytokines and express high levels of inhibitory molecules (**right**). (**B**) Focusing on the different T helper (Th) subsets, the frequency of T helper 1 (Th1) and T follicular helper (Tfh) cells is decreased in cHBV compared to acute HBV and they produce less of their signature cytokines, while the frequency of T helper 2 (Th2), T helper 17 (Th17) and regulatory Tregs are increased (depicted in red).

**Figure 3 cells-10-01114-f003:**
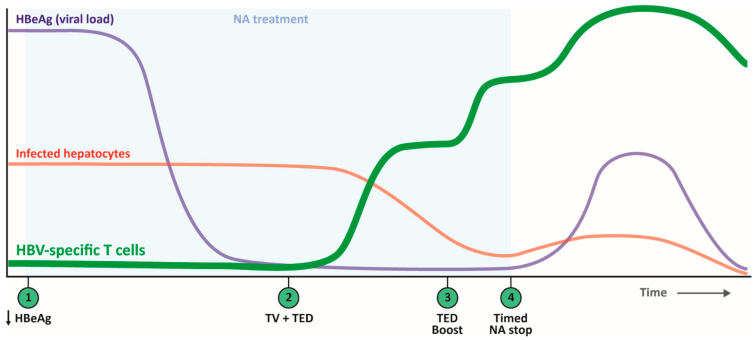
Proposed stepwise therapeutic strategy focused on opportunities to include CD4^+^ T cells. To induce multi-specific CD8^+^ and Th1 CD4^+^ T cells, first, HBeAg levels need to be reduced to remove the Th2 skewing environment by nucleos(t)ide analogue (NA) and/or siRNA treatment (step 1). When HBeAg/viral load are low, HBV-specific CD4^+^ and CD8^+^ T cells can be induced/boosted using therapeutic vaccines (TV) such as synthetic long peptide (SLP) or DNA/RNA vaccines including a Th1 skewing/IL-12 inducing adjuvant (step 2), which will result in a decrease in infected hepatocytes. TV can be combined with T cell enhancing drugs (TED) that are administered before, simultaneously or after TV depending on the type of TED (steps 2 and 3). Thereafter, when high T cell numbers of sufficient quality are obtained, a well-timed NA stop could serve as a natural booster by increasing viral load (step 4). This step-wise strategy may ultimately result in clearance of infected hepatocytes by the HBV-specific T cells and a functional cure.

## Data Availability

No new data were created or analyzed in this study. Data sharing is not applicable to this article.

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
