# Peer review of "CD4+ T Cells in Chronic Hepatitis B and T Cell-Directed Immunotherapy"

_cells, 2021, doi:10.3390/cells10051114_

Round 1
Reviewer 1 Report
The manuscript entitles “CD4+ T cells in chronic HBV and T cell-directed HBV immunotherapy” by Buschow and Jansen provide a nice overview of the importance of CD4 T cells in HBV infection as well as challenges and opportunities for therapeutic vaccination and other interventions for people with cHBV. Overall, the manuscript is well written and easy to read. The figures complement very nicely the points the researchers are trying to make regarding CD4 T cells and HBV. I believe the manuscript is ready for publication in the current format.
Author Response
We thank the reviewer for the kind words.
Reviewer 2 Report
This manuscript reviewed CD4+ T cells in chronic HBV and T cell-directed HBV immunotherapy. Buschow and Jansen described roles of CD4+ T cells in chronic hepatitis and T-cell directed immunotherapeutic strategies. However, sections 2-4 of manuscript are a summary of several previous reviews; and sections 5-6 of manuscript are a summary of the recent review by the authors’ group. Figure 1 looks very much like figure 3 in Borst et al., (2018) (ref. 23) and figure 3 is similar to figure 1 in their recent review (ref. 19). If these figures were prepared based on the previous publications, authors should acknowledge and cite these references in the figures. However, no references in the figures were citied.
Comments
References cited in lines 20, 26, 29, 39,48, 70, 72, 73-85, 91, 95-100, 106, 144, and 238 need to be replaced with references from the original observations.
Missing references in lines 67, 188, 189, 222, and 262.
In line 95, authors describe findings on the broad range of antibody subtypes in Th1 and Th2 in patients with antiviral immunity as the recent studies. However, these references were published in 2011, 1987, and 2000.
Figure 1 is similar to figure 3 of Borst et al., (2018) (ref. 23). Box caption of B cells in figure 1 is identical to figure 2 of Swain et al. (2012) (ref. 22). Figure 3 is very similar to Figure 1 of Jansen et al., (2021) (ref. 19).
Line 161, Meaning of the acute clearing infection is not clear.
Line 162, A sentence with “Furthermore and also similar to” needs to be rewritten.
In lines 185-187, authors described that a skewed Th1/Th2 ratio toward to Th2 is unfavorable for clearance of viral infection. Then, skewing Th responses into the Th2 direction could therefore be a successful immune evasion strategy for a virus. These descriptions are confusing and need to be rewritten.
Author Response
This manuscript reviewed CD4+ T cells in chronic HBV and T cell-directed HBV immunotherapy. Buschow and Jansen described roles of CD4+ T cells in chronic hepatitis and T-cell directed immunotherapeutic strategies. However, sections 2-4 of manuscript are a summary of several previous reviews; and sections 5-6 of manuscript are a summary of the recent review by the authors’ group. Figure 1 looks very much like figure 3 in Borst et al., (2018) (ref. 23) and figure 3 is similar to figure 1 in their recent review (ref. 19). If these figures were prepared based on the previous publications, authors should acknowledge and cite these references in the figures. However, no references in the figures were citied.
Authors’ response
We thank the reviewer for the thorough read of our manuscript. Our manuscript starts with an introduction on HBV and the role of CD4+ T cells in viral infections (including a supporting overview figure (Figure1)) to inform the reader and to be able to understand the real focus of our review; the role of CD4+ T cells in chronic HBV. These introductory paragraphs thus summarize broad common knowledge which has been excellently reviewed elsewhere and these reviews are also cited accordingly. We believe that apart from introductory paragraph 1 and 2, we have gathered, combined and integrated original data in a way that has not been performed before in other reviews. This resulted in a new and unique perspective focusing on the partly neglected CD4+ T cells in HBV. We like to refer to our response below regarding specific comments on the mentioned Figures 1 and 3.
Comments
Point 1: References cited in lines 20, 26, 29, 39,48, 70, 72, 73-85, 91, 95-100, 106, 144, and 238 need to be replaced with references from the original observations.
Authors’ response 1
The focus of our manuscript is the role of CD4+ T cells in (chronic) HBV. To allow the reader to fully appreciate research in this area, we started with an introduction on HBV and the role of CD4+ T cells in viral infection in general. These introductory paragraphs are broad common knowledge that have been reviewed elsewhere. We cite these reviews and believe that it is not necessary and even undesirable to include all original observations on these topics as the reference list would become too extensive and because it is not the focus of our review. To better indicate that we cited review articles in this part we have made textual adjustments in line 66, 71, 76-77 and 245-246. We have also added the most important studies on the role of CD40-CD40L interactions and cDC1 in transferring CD4 help to CD8+ T cells (line 73 and 77).
Point 2: Missing references in lines 67, 188, 189, 222, and 262.
Authors’ response 2
We thank the reviewer for pointing out that we have missed a few references. References have been added to line 69, 192 and 195. In line 222 (now 227) and 262 (now 268) we could not detect missing references as we cite original research articles in these lines.
Point 3: In line 95, authors describe findings on the broad range of antibody subtypes in Th1 and Th2 in patients with antiviral immunity as the recent studies. However, these references were published in 2011, 1987, and 2000.
Authors’ response 3
We thank the reviewer for this observation and removed ‘recent’ from the sentence and replaced it with ‘several’ (page 2, line 97).
Point 4: Figure 1 is similar to figure 3 of Borst et al., (2018) (ref. 23). Box caption of B cells in figure 1 is identical to figure 2 of Swain et al. (2012) (ref. 22). Figure 3 is very similar to Figure 1 of Jansen et al., (2021) (ref. 19).
Authors’ response 4
Figure 1 serves as an overview of the role of CD4+ T cells in the adaptive immune response to support paragraph 2 and provide the reader with an integrated extract of common knowledge to be able to understand the following paragraphs. The figure indeed contains information also outlined in Borst et al. and Swain et al. Both reviews highlight aspects important to bring to the attention of the reader before turning to HBV, the focus of our review. To prevent any misperception, we added the following sentence to the figure legend ‘This figure is an integrated extract of the most important roles of CD4+ T cells in the adaptive immune response as reviewed in Swain et al. and Borst et al.’ In addition, we adjusted the order and color distribution of the figure to reduce the similarity to the figures mentioned by the reviewer.
Figure 3 is indeed comparable in design to Figure 1 of Jansen et al., however, the focus and explanation is very different, only the timing aspect is present in both. Figure 1 of Jansen et al. is focused on the next generation therapeutic vaccines in a clinical setting including viral infection and immune monitoring. Figure 3 depicts similar timing, however, it is focused on (CD4+) T cells and how to increase their numbers and quality and is based on T cell-focused supporting literature. Therefore, we believe Figure 3 is a different figure from Figure 1 of Jansen et al. Nonetheless, to highlight the focus on T cells, we have made adjustments to the color scheme of the figure.
Point 5: Line 161, Meaning of the acute clearing infection is not clear.
Authors’ response 5
We thank the reviewer for indicating that the meaning of acute infection in line 161 is unclear. We have adjusted the sentence to the following ‘So, in acute self-limiting infection a strong, multi-specific Th1…. ‘ (page 4, line 163).
Point 6: Line 162, A sentence with “Furthermore and also similar to” needs to be rewritten.
Authors’ response 6
We have adjusted the sentence to the following ‘Furthermore (and also similar to HBV-specific CD8+ T cells), HBV-specific CD4+ T cells….’ (page 4, line 168).
Point 7: In lines 185-187, authors described that a skewed Th1/Th2 ratio toward to Th2 is unfavorable for clearance of viral infection. Then, skewing Th responses into the Th2 direction could therefore be a successful immune evasion strategy for a virus. These descriptions are confusing and need to be rewritten.
Authors’ response 7
We thank the reviewer for pointing out this unclarity. We have adjusted lines 191-194 to the following ‘A skewed Th1/Th2 ratio towards Th2 cells is considered unfavorable for clearance of viral infections. Skewing Th responses into the Th2 direction could therefore be a successful immune evasion strategy for a virus as this will lead to a Th response that is not capable of clearing the virus, giving the virus a survival advantage.’
Reviewer 3 Report
General comment
The authors have compiled an excellent review on a subject which has indeed been a bit neglected compared to the role of CD8 lymphocytes. HBV has evolved a plethora of properties which undermine its effective immune control. As the authors justly point out, CD4 lymphocytes may be the key element to strengthen the weak immune response against HBV. The text is well written, the references are well selected and commented in a way that a coherent picture is drawn. The figures are well designed and instructive. One aspect should possibly be discussed in a bit more detail: the reactivation of anergic B cells.
Some minor points to correct:
- Avoid abbreviation of HBV in title.
- Correct: … leading to acute or persistent liver infection …
- L31, 32. Here are some words are missing. Most patients start with the noninflammatory phase and some remain in this phase.
- L23, 47. Use either the terms envelope or surface but not both. I suggest surface because of HBsAg.
- L78, Explain the abbrevation TCR here.
- Extent, not extend.
- Replace anti-HBsAg and anti-HBeAg by the usual terminology: anti-HBs and anti-HBe.
- … in, not is
- … genetically encoded HBV antigens … is not a good terminology. I suggest … nucleic acids encoding HBV antigens … .
- What does “whole protein HBsAg” mean? The small or all three HBs proteins?
- I presume, preS should preS2.
Author Response
The authors have compiled an excellent review on a subject which has indeed been a bit neglected compared to the role of CD8 lymphocytes. HBV has evolved a plethora of properties which undermine its effective immune control. As the authors justly point out, CD4 lymphocytes may be the key element to strengthen the weak immune response against HBV. The text is well written, the references are well selected and commented in a way that a coherent picture is drawn. The figures are well designed and instructive. One aspect should possibly be discussed in a bit more detail: the reactivation of anergic B cells.
Authors’ response
We thank the reviewer for the kind words. In this review we focus on T cells and the help of CD4+ T cells to CD8+ T cells, but nonetheless we agree that reactivation of anergic B cells is worth to mention. Therefore, we added the following words to line 298-300 on page 7 ‘in addition to reactivation of anergic HBV-specific B cells.’ We also added ‘together with neutralizing antibodies’ to line 346 on page 8.
Some minor points to correct:
Point 1: Avoid abbreviation of HBV in title.
Authors’ response 1
We have adjusted the title to the following ‘CD4+ T cells in chronic hepatitis B and T cell-directed immunotherapy’ (page 1, line 2-3).
Point 2: Correct: … leading to acute or persistent liver infection …
Authors’ response 2
We have corrected the sentence according to the reviewers suggestion ‘The hepatitis B virus (HBV) specifically infects hepatocytes leading to acute or persistent liver infection (page 1, line 18-19).
Point 3: L31, 32. Here are some words are missing. Most patients start with the noninflammatory phase and some remain in this phase.
Authors’ response 3
We have modified the sentence to the following ‘Most patients start with a non-inflammatory phase with high levels of HBV DNA and HBsAg, presence of HBeAg and normal ALT levels (HBeAg-positive chronic infection; EPCI), followed by an inflammatory phase with increased ALT levels, fluctuating HBV DNA levels and the presence of HBeAg (HBeAg-positive chronic hepatitis; EPCH), while some patients remain in the EPCI phase. The EPCH phase may mount into partial viral immune control with loss of HBeAg expression, often accompanied by generation of antibody responses to HBeAg, undetectable or low HBV DNA levels and normalization of ALT levels (HBeAg-negative chronic infection; ENCI). (page 1, line 31-38).
Point 4: L23, 47. Use either the terms envelope or surface but not both. I suggest surface because of HBsAg.
Authors’ response 4
We thank the reviewer for pointing out this inconsistency. We have adjusted the term envelope to surface according to the reviewer’s suggestion (page 2, line 48; page 4 line 150-151 and page 7 line 267).
Point 5: L78, Explain the abbrevation TCR here.
Authors’ response 5
We have added the explanation of the abbreviation TCR to line 80-81 on page 3.
Point 6: Extent, not extend.
Authors’ response 6
We have replaced extend with extent in line 94 on page 2.
Point 7: Replace anti-HBsAg and anti-HBeAg by the usual terminology: anti-HBs and anti-HBe.
Authors’ response 7
We have replaced anti-HBsAg and anti-HBeAg with anti-HBs and anti-HBe respectively (page 6, line 228 and page 7, line 274).
Point 8: … in, not is
Authors’ response 8
We have replaced ‘is’ with ‘in’ in line 238 on page 6.
Point 9: … genetically encoded HBV antigens … is not a good terminology. I suggest … nucleic acids encoding HBV antigens … .
Authors’ response 9
We have replaced ‘genetically encoded’ with ‘nucleic acids encoding’ (page 6, line 249-250).
Point 10: What does “whole protein HBsAg” mean? The small or all three HBs proteins?
Authors’ response 10
We thank the reviewer for this relevant question and removed ‘whole protein’ from the sentence (line 261 page 7) In Ren et al. a vaccine composed of S combined with a small amount of pre-S2 was used, while in Couillin et al. a vaccine composed of pre-S2 was used, so using the term HBsAg will cover both studies.
Point 11: I presume, preS should preS2.
Authors’ response 11
We thank the reviewer for pointing out this mistake. PreS should indeed be preS2. We replaced pre-S with pre-S2 in line 264 on page 7.
Round 2
Reviewer 2 Report
All the comments has been addressed.